# Higher Rates of Viral Evolution in Chronic Hepatitis B Patients Linked to Predicted T Cell Epitopes

**DOI:** 10.3390/v17050684

**Published:** 2025-05-08

**Authors:** Magnus Illum Dalegaard, Anni Winckelmann, Ulrik Fahnøe, Alexander P. Underwood, Anders Gorm Pedersen, Signe Bollerup, Jens Bukh, Nina Weis

**Affiliations:** 1Department of Infectious Diseases, Copenhagen University Hospital, 2650 Hvidovre, Denmark; magnus_dalegaard@hotmail.com (M.I.D.); anni.assing.winckelmann@regionh.dk (A.W.); ulrik@sund.ku.dk (U.F.); alexander.underwood@sund.ku.dk (A.P.U.); signe.bollerup@regionh.dk (S.B.); jbukh@sund.ku.dk (J.B.); 2Copenhagen Hepatitis C Program (COHEP), Department of Infectious Diseases, Copenhagen University Hospital, 2650 Hvidovre, Denmark; 3Department of Immunology and Microbiology, Faculty of Health and Medical Sciences, University of Copenhagen, Blegdamsvej 3B, 2200 Copenhagen, Denmark; 4Department of Health Technology, Section for Bioinformatics, Technical University of Denmark, 2800 Kongens Lyngby, Denmark; 5Department of Clinical Medicine, Faculty of Health and Medical Sciences, University of Copenhagen, 2300 Copenhagen, Denmark

**Keywords:** HBV, next-generation sequencing, T cell epitopes, mutation rate, DANHEP

## Abstract

The impact of hepatitis B virus (HBV) diversity and evolution on disease progression is not well-understood. This study aims to compare intra-individual viral evolution in two groups of chronic hepatitis B (CHB) patients, using antiviral treatment initiation as a measure of lack of immunological control. From the Danish Database for Hepatitis B and C (DANHEP), 25 CHB patients were included; 14 with antiviral treatment initiation (TI group), and 11 without (NTI group). For each patient, three serial plasma samples taken before potential treatment initiation were selected. HBV DNA was amplified by PCR and analyzed by next-generation sequencing. HBV DNA and alanine transaminase were elevated in the TI group throughout the study period. Significantly higher substitution rates in the NTI group versus the TI group were found both within the viral population and at consensus level. Putative predicted CD8^+^ T cell epitopes contained significantly more substitutions in the NTI group. Genome-wide association analysis revealed several amino acid residues in the HBV genome associated with treatment initiation. This study shows that HBV has a higher rate of substitutions in CHB patients not requiring treatment. This could be linked to host immune pressure leading to disease control.

## 1. Introduction

Hepatitis B virus (HBV) infection can lead to chronic hepatitis B (CHB), which increases morbidity and mortality due to the risk of development of liver cirrhosis and hepatocellular carcinoma (HCC) [1]. Approximately 254 million individuals were predicted to be infected with HBV worldwide, in 2022, with an estimated 1.1 million deaths annually [2]. While no cure exists, HBV can be supressed with nucleotide analogues (NAs) [3,4]. HBV DNA levels are the strongest predictive biomarker associated with disease progression, and viral suppression is associated with a reduced incidence of cirrhosis and HCC, as well as cirrhosis regression [5,6,7]. However, not all CHB patients are candidates for NA treatment, which is initiated based on predictors of disease progression, such as elevated alanine transaminase (ALT), elevated HBV DNA, development of fibrosis/cirrhosis, immunosuppression, family history of HCC, or pregnancy with HBV DNA above 200,000 IU/mL [6].

HBV, a DNA virus of ~3.2 kilobases (kbs), is a member of the *Hepadnaviridae* family, and is categorized into 10 genotypes (A–J), with >8% sequence diversity between genotypes. The genotypes are associated with different clinical manifestations, routes of transmission, and global distribution [7,8]. The HBV genome consists of four open reading frames (ORFs), of which some overlap leading to single mutations potentially affecting more than one of the seven encoded proteins: Hepatitis B core antigen (HBcAg), HBV polymerase/reverse transcriptase (RT), Hepatitis B e antigen (HBeAg), Hepatitis B surface antigen (HBsAg) (which consists of three proteins: large, middle, and small), and the Hepatitis B X protein (HBx) [6].

The HBV virion contains partially double-stranded, relaxed, circular DNA (rcDNA) covalently linked to the HBV polymerase. The virion infects hepatocytes, where the rcDNA is transported to the nucleus and converted into double-stranded DNA, which is ligated into a covalently closed circular DNA (cccDNA). cccDNA is highly stable and stays in the hepatocytes after clearance of the circulating virus [5]. The cccDNA serves as a template for transcription of several species of RNAs including pregenomic RNA (pgRNA). During replication, pgRNA serves as a template for reverse transcription to the negative-strand DNA and subsequently the positive-strand synthesis. Due to the lack of proof-reading activity in the reverse transcription, replication leads to an increased rate of mutations. Therefore, intra-host heterogenous HBV species can be found in infected individuals [6,9].

HBeAg status is correlated with HBV replication and infectivity [10]. HBeAg-absence and detection of antibodies against HBeAg (anti-HBe) is favourable in CHB infection and often represent immunological control [6,9]. The ultimate endpoint of CHB therapy is loss of HBsAg, which reduces the risk of HCC and is considered a functional cure.

The innate and adaptive immune system plays a significant role during HBV-infection through mechanisms involving B and T cell responses. The virus has, however, developed mechanisms to escape the immune system and the interaction between the host and HBV determines the outcome of infection [10,11].

At present, there are no tools to predict the risk of disease progression for CHB patients, and therefore, CHB patients are monitored bi-annually. It is known that certain mutations in the HBV genome are associated with clinical outcomes, such as immune escape, HBeAg negativity, and HCC development [12]. Mapping potential viral factors associated with disease progression could allow for stratification of patients with low versus high-risk of progression. This would be beneficial for both the individual and the society by reducing frequency of hospital follow-ups for low-risk individuals.

The objective of this study was to compare intra-individual HBV evolution over time in two groups of CHB patients, using treatment initiation as an indicator of immunological control failure.

## 2. Materials and Methods

### 2.1. The Danish Database for Hepatitis B and C (DANHEP) and Associated Biobank

Samples for this study were selected from the Danish Database for Hepatitis B and C (DANHEP) biobank. DANHEP is a nationwide, prospective cohort study, with ongoing enrolment of patients over the age of 15 years with in- or out-patient hospital visits due to CHB and/or chronic hepatitis C (clinically defined as viremia for longer than six months) in one of 16 departments across Denmark. Upon enrolment and during follow-up for CHB patients, data such as sex, age, country of origin, infecting agent, viral genotype, mode of transmission, viral load, serology, ALT, fibrosis assessed by transient elastography (TE) and/or liver biopsy, and co-infection with hepatitis C virus (HCV) or human immunodeficiency virus (HIV) are registered. Furthermore, DANHEP contains information about treatment indication, drug initiation and cessation dates, and effect of treatment. For this study on CHB, data regarding sex, age, country of origin, HBV DNA, HBeAg, anti-HBe, genotype, antiviral treatment (date of treatment initiation and nucleoside analogue), treatment indication, and HCV or HIV co-infection were extracted. A biobank is connected to DANHEP, where annual blood samples from the included patients are stored, and where the samples used in this study were collected.

### 2.2. Study Subjects

Inclusion criteria for this study were patients with CHB included in DANHEP, who were divided into either a treatment initiation (TI) group or a non-treatment initiation (NTI) group, with a minimum of three available samples for analysis. The patients in the TI group were treated with Tenofovir or Entecavir.

Plasma samples stored at −80 degrees Celsius (°C) had been collected before possible antiviral treatment initiation at three time points: A (at inclusion in DANHEP), B (as close to midpoint enrolment as possible), and C (the latest sample available, for the TI group before treatment initiation). In the NTI group, a lower limit for HBV DNA of 10,000 IU/mL was set to account for sequencing assay sensitivity. For the TI group, a lower limit for HBV DNA of 100,000 IU/mL was set, to differentiate them further from the NTI group. Exclusion criteria were co-infection with HCV or HIV.

The patients also had peripheral blood mononuclear cells stored at −80 °C as a part of the TANDEM HBV cohort.

### 2.3. DNA Extraction and Amplification

DNA was extracted from 500 µL plasma by using the QIAamp^®^ UltraSens^®^ Virus Kit (QIAGEN, Venlo, The Netherlands), according to the manufacturer’s instructions. The eluted DNA was amplified by PCR using Q5^®^ Hot Start High-Fidelity 2X Master Mix (New England Biolabs, Ipswich, MA, USA). Three amplicons which together covered the full-length genome were obtained using three pairs of primers: 1st forward primer 5′ TTTTTCACCTCTGCCTAATCATCTCTTG 3′; 1st reverse primer 5′ GTAGGCTGCCTTCCTGACTG 3′; 2nd forward primer 5′ GGGTCACCATATTCTTGGGAACA 3′; 2nd reverse primer 5′ CGAACCACTGAACAAATGGCACT 3′; 3rd forward primer 5′ TGCACCTGTATTCCCATCCCA 3′, and 3rd reverse primer 5′ AAAAAGTTGCATGGTGCTGGTGC 3′.

The samples were placed in a thermocycler at 98 °C for 30 s, followed by 35 cycles of: 98 °C for 10 s, 65 °C for 10 s, 72 °C for two minutes, and 72 °C for three minutes. The PCR product was purified using a DNA Clean and Concentrator^®^ -5 kit (Zymo Research, Irvine, CA, USA) according to the manufacturer’s instructions. The DNA concentration was measured using a NanoDrop^®^ (Thermo Fisher Scientific, Waltham, MA, USA). Afterwards seventeen nanograms (ng) of HBV DNA from each of the three part-length sequences were pooled to reach a total of 50 ng PCR product and diluted to a final concentration of 2.5 ng/µL to be used in library preparations.

### 2.4. Sequencing and Analysis

Library preparations were performed on the PCR product pool using a NEBNext Ultra II FS DNA library prep kit (New England Biolabs) according to the manufacturer’s instructions. The preps were purified using AMPure XP beads and sequenced on a Miseq using 2 × 300 pair-end read configuration. Sequence analysis was performed as previously described [13]. Consensus sequences were de-novo assembled by the Iterative Virus Assembler (IVA) tool. Reads were mapped using Burrows-Wheeler Aligner (BWA) tool to further refine the consensus and to compare later samples to the first sample. Reads were preprocessed by removing primer 5′ anchor sequences by cut-adapt. Sickle was used to trim, and filter reads with a cut-off of 30. A coverage threshold of 100 reads was set to properly call consensus. Samples below that threshold were excluded from further analysis.

Consensus sequences were aligned using the software MAFFT (V7.490), and phylogenetic trees were reconstructed using PhyML (V20120412) and visualized in FigTree (V1.4.3). Sequences were compared using Geneious (version 2023.0.4), with a substitution matrix extracted from the alignment. Substitution rates were estimated using linear regression in Microsoft Excel (v16.63). Figures were made using GraphPad Prism (version 9.4.1).

### 2.5. Statistical Analysis

Comparisons between the TI and NTI groups were done using Mann Whitney U tests (age, ALT, HBV DNA), Fischer’s exact tests (sex, country of origin, genotype, HBeAg, anti-HBe, and TE status at baseline), and Welsh’s Two-Sample *t*-tests (follow-up duration). All tests were two-sided, and statistical significance was defined as a *p* value < 0.05. Categorical variables were summarized with frequency and proportion (*n*, %), and continuous variables with median (range). All tests were performed using R studio (version 2022.02.01).

### 2.6. Synonymous and Non-Synonymous Mutations

To investigate mutations in the HBV genome over time, single nucleotide polymorphism (SNP) genie [14] was used to estimate within-pool πN/πS and dN/dS [14,15]. The ORFs core, pol, and S were predicted using transdecoder [14] and subsequently checked manually. Time points B and C were analyzed compared to the A consensus sequence for each ORF using a sliding window of five codons.

### 2.7. CD8^+^ T Cell Epitope Analysis

Patients were typed for human leukocyte antigen (HLA) using a protocol previously described [16,17]. In brief, patient DNA was obtained by extracting DNA from the PBMCs using a DNeasy Blood and Tissue kit (QIAGEN, Venlo, The Netherlands), according to the manufacturer’s instructions. PCR amplification of A, B, and C HLA alleles from the patients were performed using the primers and PCR cycling conditions previously described [16]. AMPure XP beads (Qiagen) were used to purify the products according to the manufacturer’s instructions. Library preparation was conducted on the pooled purified products using a NEBNext Ultra II DNA Library Preparation kit (New England Biolabs, Ipswich, MA, USA) and sequenced by NGS using the MiSeq platform (Illumina, San Diego, CA, USA). Subsequently, the data were analyzed using Hisat-genotype analysis pipeline, as previously described [18]. Using the HBV sequences obtained above, the core protein, PreS protein, polymerase, and X protein were extracted using Geneious (software version 2023.0.4) and submitted to the MHC class I T cell epitope prediction tool via the Immune Epitope Database (IEDB). Each protein from each time point was submitted with the patient’s corresponding HLA type to identify potential CD8^+^ T cell epitopes. Predicted epitopes were selected based on a processing score >1.5, an MCH binding IC_50_ <100 (MHC score > −2.00), and a total score >0. Predicted epitopes reaching these criteria were then compared longitudinally between time points A, B, and C, and substitutions in these epitopes were reported.

### 2.8. Genome-Wide Association Studies of Amino Acid Variants Associated with Patient Status

The software BMAGWA version 2 [19] was used to search for mutations associated with whether a patient received treatment for HBV. BMAGWA is a tool for genome-wide association studies (GWAS). In this context, a binary phenotype (TI group or NTI group) was used, while amino acid alignments of viral sequences from patients were used as pseudo-genotypes. These were converted to plink-format using in-house scripts and software, as previously described [20]. One important feature of this software is that instead of investigating variable sequence sites individually (as is usually done in GWAS), it simultaneously considers all variants for inclusion as additive predictors of the phenotype. The number of sites involved is regularized by setting a prior distribution on the expected number (10 +/− 6 in this case), and the software then searches the full space of possible additive combinations of sites that could have a joint impact on the phenotype. Specifically, all variable sites in the amino acid alignments of the four concatenated proteins were extracted and encoded as diploid (DNA) genotypes: If the reference (most common) amino acid was present at a site, it was encoded as A/A, while any non-reference amino acid was encoded as G/G. BMAGWA was run with the following settings: Number of iterations: 10^7^; number of chains: 5; warm-up: 50% of iterations; expected number of sites with an impact on phenotype (e_qq): 10; variance on that parameter (var_qq): 10; number of samples saved: 10,000 per chain; genetic model: dominant. The output from BMAGWA was postprocessed using in-house software to find the posterior probability that a site has an impact on the need for treatment, and to compute the corresponding Bayes factor (BF—the ratio between posterior and prior odds). To help distinguish between real and spurious signals, the BMAGWA analysis was performed on 1000 data sets where the phenotype vector had been randomly shuffled and then focused on those sites where BF > 3, and BF was larger than the BF for 95% of the shuffled data sets.

## 3. Results

### 3.1. Patient Characteristics

As shown in Figure 1, 25 patients met the inclusion criteria and were included: 14 patients in the TI group and 11 in the NTI group. The complete collection of samples (A, B, and C) spanned from 10 to 112 months (median = 60 months), with no significant difference in follow-up time between the groups (*p* = 0.70). Among the patients in the TI group, 13 patients were treated with Tenofovir, and one with Entecavir.

Following inclusion in the study, HBV DNA and ALT results were obtained from the patients’ electronic files, as opposed to data registered in DANHEP, as some patients did not have corresponding HBV DNA values registered in the database for all three sample time points (A-C). The obtained data showed that many of the obtained patient samples had corresponding HBV DNA values < 10,000 IU/mL. Furthermore, two patients in the NTI group had one sample at time point C, with corresponding HBV DNA > 100,000 IU/mL (Patient 05 and 13; HBV DNA 12,900,000 and 175,000 IU/mL, respectively) exceeding the level of viremia set at inclusion for patients in the NTI group.

Baseline patient characteristics are presented in Table 1. There were no significant differences found in sex, age, follow up duration or country of origin between the two groups. However, the TI group had significantly higher levels of HBV DNA and ALT throughout the study period.

At time point A, significantly more patients were HBeAg positive (*p* = 0.001) and anti-HBe negative (*p* = 0.0078) in the TI group vs. the NTI group (64% vs. 0% and 50% vs. 0%, respectively). Furthermore, there were significantly more patients with TE >7 kPa, indicating fibrosis in the TI group (*p* = 0.015).

### 3.2. Sequencing Results

Samples that met the coverage threshold, with full HBV genome sequences being obtained were included in further analysis. In the TI group, 13 patients had all three samples included, while one patient had two samples included. In the NTI group, four patients had three samples included, three patients had two samples included, one patient had one sample included, and three patients had zero samples included in the study.

A phylogenetic analysis was performed using obtained full HBV genome sequences [21,22,23,24] (Figure 2). The phylogenetic tree shows the genotype distribution of the 21 patients with ≥ 2 samples included in the sequence analyses; sequences from individual patients grouped together. One patient was found to be infected with genotype A, seven patients with genotype B, seven patients with genotype C, five patients with genotype D, and one patient with genotype E, representing the heterogenetic distribution of HBV genotypes in Denmark [25]. No significant differences in the distribution of genotypes between the two groups of patients were detected among the 21 patients (*p* = 0.33).

### 3.3. Substitution Rates

Figure 3A shows combined substitution rates, reported as number of substitutions per site per year, calculated using linear regression from sample A to sample B and from sample A to sample C. For the one patient in the NTI group, where sample A was excluded (patient 05), the substitution rates from B to C were calculated.

Substitution rates ranged from 0 to 3.60 × 10^−3^ with a median of 6.41 × 10^−4^ in the TI group and 1.62 × 10^−3^ in the NTI group (Figure 3A). Patients in the TI group had significantly lower substitution rates compared to the NTI group (*p* = 0.03, Mann Whitney U test, Figure 3A). When not stratified by treatment-initiation, a significant difference in substitution rates between patients with HBeAg negativity and positivity at time point A was found, with patients being HBeAg negative having higher rates (*p* = 0.04) (Figure 3B). Through linear regression, a significant association between high substitution rates and low levels of HBV DNA was found (*p* = 0.0065, r^2^ = 0.3297) (Figure 3C).

### 3.4. Intra-Patient Viral Evolution

The intra-patient sequence heterogeneity was analyzed in the Core, S, and Pol reading frames to further investigate the significance of the detected differences in mutation rates. The mean number of non-synonymous differences from the reference per non-synonymous site (dN) and the mean number of synonymous differences from the reference per synonymous site (dS) was analyzed using the time point A consensus sequence as a reference. Comparison of dN and dS between the two groups revealed a trend towards higher dN and dS values in the NTI group compared to the TI group in all three ORFs at time points A and C (Figure 4A–D). These findings were confirmed by comparing the mean number of pairwise non-synonymous differences per non-synonymous site (πN), as well as mean number of pairwise synonymous differences per synonymous site (πS) in a sliding window analysis where higher levels in the NTI group were found when compared to the TI group (Figure 4E,F). The πN and πS graphs revealed areas of the proteins in both groups to be conserved while other areas with potential positive selection, could be adaptations of the virus to evade the immune pressure. To further investigate transition and transversion frequencies that could be a possible effect of APOBEC cytidine deaminases, defined by C to T and G to A transitions, mutational analysis of the SNPs was performed (Appendix A). No clear difference was observed in the overall frequency between the transitions C to T and G to A, compared to T to C and A to G. However, a bigger spread between the individual patients could be observed for C to T and especially G to A, which could be a result of APOBEC cytidine deaminases mutagenesis in some individuals.

### 3.5. Substitutions in Predicted Epitopes

For 19 patients with sequence data (6 in the NTI group and 13 in the TI group), HLA typing was performed to identify the HLA A, B, and C alleles (Appendix A). By comparing predicted CD8^+^ T cell epitopes across the three different time points (A-C) for each patient, several possible escape substitutions in the predicted CD8^+^ T cell epitopes were found (Appendix A). The NTI group was found to have a significantly higher number of altered CD8^+^ T cell epitopes (*p* = 0.0144, Mann Whitney U test, Figure 5A), and a higher frequency of altered CD8^+^ T cell epitopes (*p* = 0.0067, Mann Whitney U test, Figure 5B). The higher rate of epitope changes could be caused by selective pressure from cellular immunity in the NTI group. In contrast, patients in the TI group exhibited a significantly lower substitution rate in predicted epitopes, which could suggest a less efficient immune response. No putative escape mutations in the predicted epitopes of the highly immunodominant region of the S-gene was found (residues 101–172) in either of the patient groups [26].

### 3.6. Search for Amino Acid Variants Associated with the Need for Treatment

A Bayesian GWAS approach was used to search for protein mutations associated with whether a patient required treatment for HBV. Specifically, all variable sites in the amino acid alignments of the four proteins: Core, Pol-trans, Pre-S, and X, were extracted and encoded based on whether a site contained the reference amino acid (the most frequent one), or not (so all non-reference amino acids are encoded identically). Using this approach, 9 high confidence sites that were associated with treatment status were found (Figure 6A). The indicated sites include one in Core, five in Pol-trans, and three in Pre-S. The amino acid mutations at Pol-trans location 582 and Pre-S location 427 are in overlapping codons and are caused by a single nucleotide change. The sequence logos in Figure 6B summarize the amino acid usage in the treatment and non-treatment groups.

## 4. Discussion

This study is, to our knowledge, the first to compare pre-treatment viral genomic evolution in CHB patients with—and without antiviral treatment initiation. The two groups differed in all parameters associated with disease progression.

For the NTI group, significantly higher overall substitution rates, and a higher rate of substitutions specifically in predicted CD8^+^ T cell epitopes were found suggesting a higher level of selective pressure from the adaptive immune system. We would presume that lower rate of substitutions in the TI group is due to a weaker cellular adaptive T cell response, which does not select for possible escape mutations in the viral CD8^+^ T cell epitopes. This could be related to T cell exhaustion for the TI group [27]. This is further supported by the consistent low levels of viremia in the NTI group and potentially highlights the importance of controlling HBV viremia via virus-specific cellular immunity. Low HBV DNA titers in the blood were found to be associated to high mutation rates. However, the patients with below 10.000 IU/mL or undetectable HBV DNA might show different mutation rates that were not evaluated in this study. Interestingly, no association was found between disease status and known antibody epitope mutations that could potentially lead to immune escape in patients experiencing flares or reactivation [28]. Thus, cellular immunity likely represents an important factor for disease progression. In contrast, low substitution rates were found to be associated with high viral loads as well as HBeAg positivity. HBeAg positivity is known to be associated with high viral replication [9], which could, in part, explain the corresponding findings between HBeAg status, high viral load, treatment initiation, and substitution rates. This study confirms previous findings that viral substitution rate is associated with HBeAg status [29,30]. Previous studies have shown similar mutation rates as this study but also a low evolutionary rate, meaning low fixation of new mutations, compared to RNA viruses with similar mutation rates [31,32]. This can be explained by the HBV cccDNA, which has a long lifespan in resident liver cells without need of replication, thereby slowing down the fixation of mutations [31]. In this study, we sequenced the circulating HBV replicating DNA, and not the cccDNA, which would require liver biopsies not available for these patients. However, future studies looking at cccDNA mutation rates will be able to shed light on the evolutionary rates and adaptation of HBV.

Genome wide association analysis (GWAS) of the four ORFs revealed residues associated with disease progression. Two were found in confirmed antibody epitopes in preS [33,34] and could be modulators of immune evasion. PreS residue 168 is situated in a described antibody epitope and two NTI group patients had mutations in that area [34]. However, 67 PreS is situated in a confirmed antibody epitope known to be specific for the adw2 serotype [33]. Nonetheless, these patients have not been serotyped, and further mutations in that epitope could still be an attempt of viral escape. The GWAS approach also identified several residues in the polymerase protein that were likely to play a role in disease progression [35]. Interestingly, the polymerase residue 37 situated in the terminal protein domain of the N-terminal helices is important for binding and packaging of the viral RNA and further synthesis of the genomic DNA [35,36]. The Core residue 64 was mutated in three NTI group patients, and this residue is linked to a T cell epitope restricted to HLA-A*02, but none of these patients were found to have this HLA allele [37]. This is, to our knowledge, the first GWAS to associate disease progression with the HBV viral sequence. In addition, the analysis is not biased by the need of prior knowledge of epitopes or functional regions and can therefore reveal new residues important for disease control and treatment initiation.

The strength of this study was the access to the DANHEP database and biobank which allowed for selection of serial pre-treatment plasma samples spanning up to 18 years and for categorization into two groups based on subsequent treatment initiation. The sequencing of full-length HBV at three time points enabled a better estimation of the mutation rate and analysis of possible mutation reversions.

A limitation of this study was that, though international clinical guidelines attempt to standardize criteria for treatment initiation in CHB patients, it remains an individual decision for clinicians. To address this limitation, it was ensured that the two groups differed in clinical aspects related to both viral activity and inflammation of the liver, such as ALT levels and fibrosis state. The two groups were however not matched by time since infection, and it is not possible to exclude future disease progression and potential antiviral treatment in the NTI group. Moreover, the study was limited by a small sample size, low sequencing assay sensitivity, and the samples available in the biobank. Even though only patients with three samples registered with HBV DNA >10,000 IU/mL were included, some patient samples had very low levels of viremia and did not yield sequencing data that could be analyzed.

## 5. Conclusions

Compared to the TI group, the NTI group displayed consistently low levels of viremia, and significantly higher substitution rates and levels of alterations in predicted CD8^+^ T cell epitopes, suggesting increased host immune pressure on the virus in this group. Amino acid residues in Core, PreS, and Pol associated with treatment and disease status were identified. This study enhances the understanding of the viral pathogenesis and the complex HBV versus host interaction. Along with in-depth sequencing analysis studies, this could possibly allow for stratification of patients into high- and low-risk groups in the future, benefiting both patients and society.

## Figures and Tables

**Figure 1 viruses-17-00684-f001:**
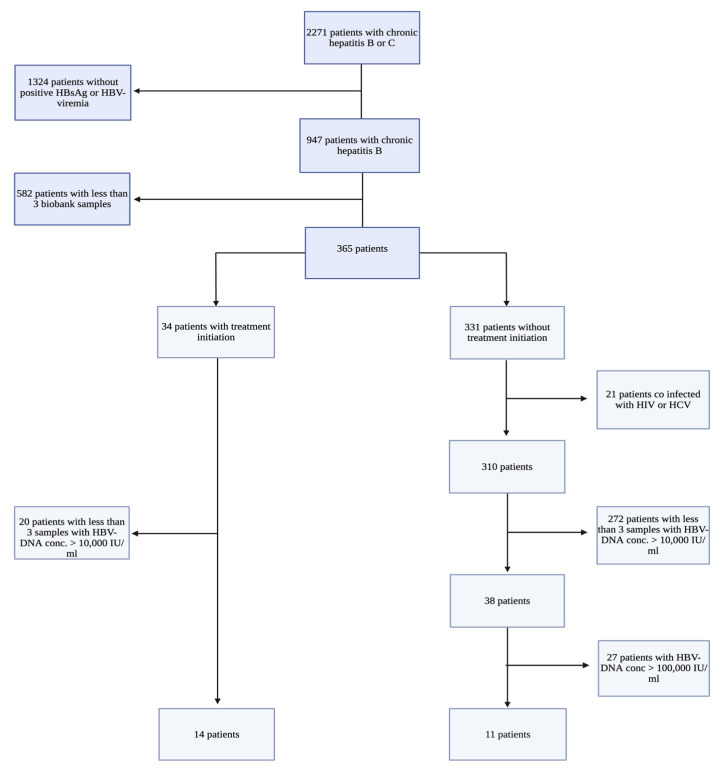
Flowchart of the selection of patients with Chronic Hepatitis B (CHB), including inclusion and exclusion criteria. All study participants were patients with CHB included in the Danish Database for Hepatitis B and C (DANHEP) at Copenhagen University Hospital, Hvidovre, Denmark, between 2002–2019. Patient samples were collected from a biobank connected to DANHEP, where annual plasma/serum samples from included CHB patients are stored. HBV DNA: Hepatitis B virus DNA measured in international units/mL (IU/mL), HBsAg: Hepatitis B surface antigen, HIV: Human Immunodeficiency virus, HCV: Hepatitis C virus. This flowchart was created using biorender.com.

**Figure 2 viruses-17-00684-f002:**
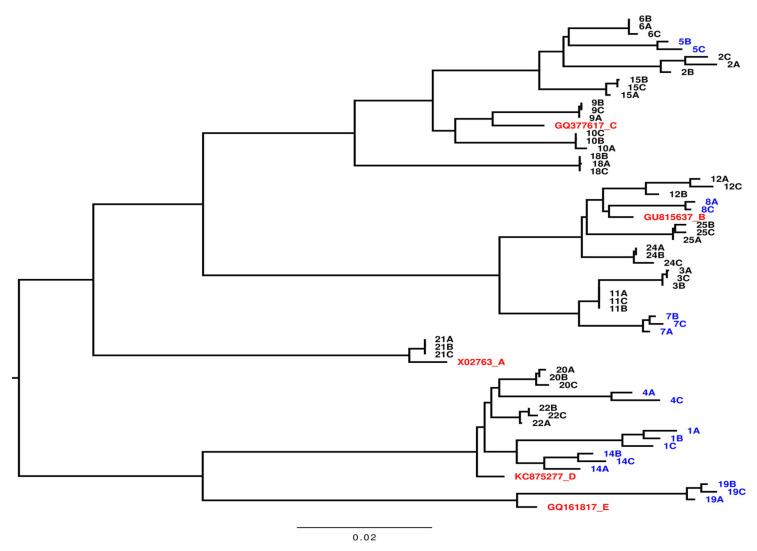
Phylogenetic tree of full-length viral genome sequences recovered from samples from 21 patients with Chronic Hepatitis B. Each number represents the individual patient number. A is the first sample taken, B is the second sample, and C is the third sample. Samples marked with blue are from 7 patients without treatment initiation (NTI), the black markings are from 14 patients with treatment initiation (TI), and the red markings are the HBV genotype references as indicated. The bar illustrates the branch length with nucleoside changes per site.

**Figure 3 viruses-17-00684-f003:**
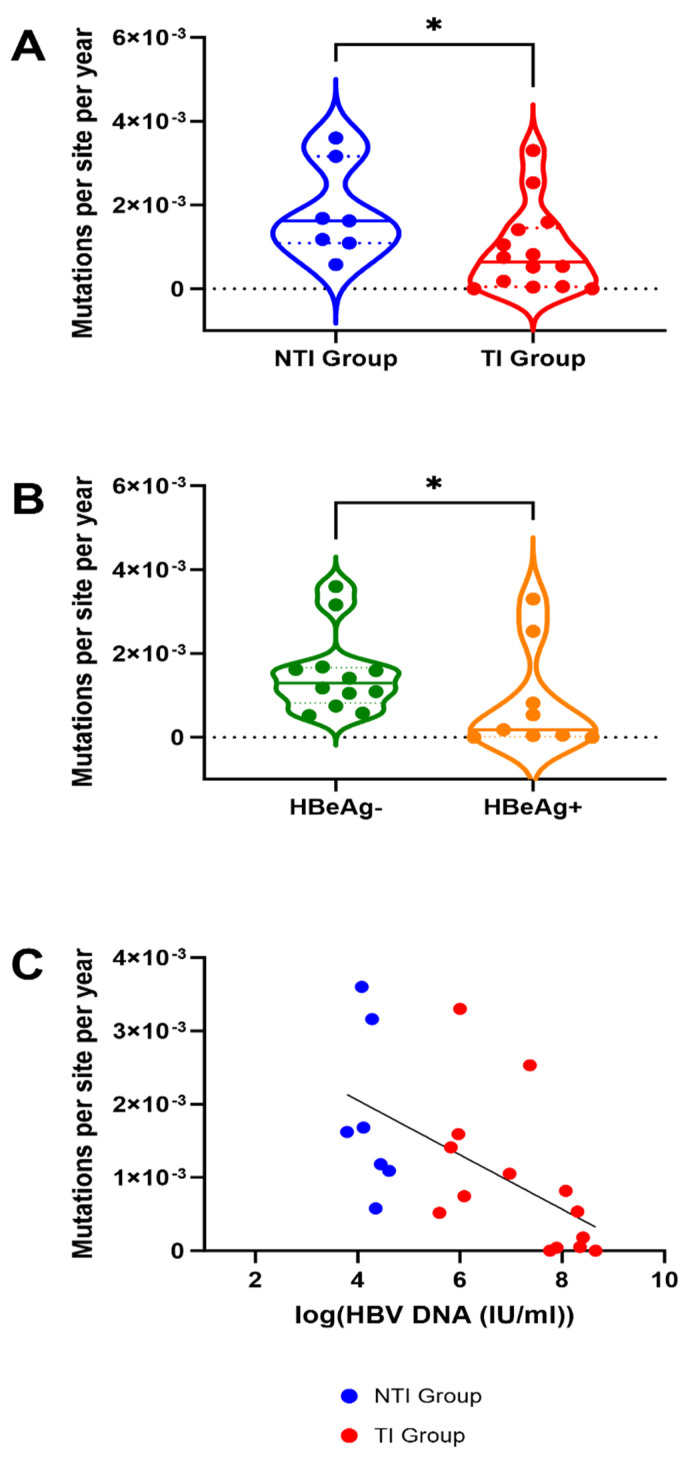
Mutation rates in 21 patients with Chronic Hepatitis B with and without treatment initiation, HBeAg negative and positive patients and association with HBV-viremia. (**A**) Substitution rates are significantly higher in the group of patients without treatment initiation (NTI) compared to those with treatment initiation (TI) (*: Mann Whitney U *p*-value < 0.005). (**B**) Substitution rates are significantly higher in patients with negative HBeAg status at baseline. For A and B, the dashed lines inside the plot represent median value, and the dotted lines represent the upper and lower quartiles. (**C**) The mutation rates are significantly correlated with HBV DNA level, the line is showing the linear regression.

**Figure 4 viruses-17-00684-f004:**
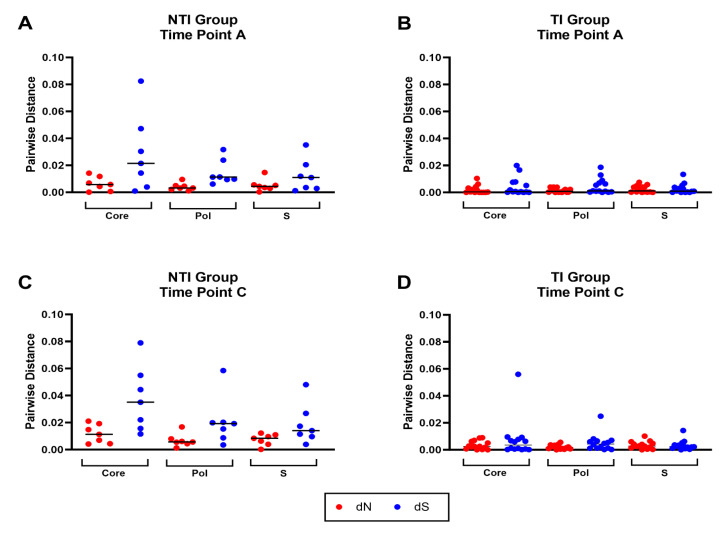
Distribution and frequency of non-synonymous and synonymous mutations in the HBV open reading frames core, pol, and S. Panel (**A**–**D**) show the mean number of non-synonymous differences from the reference per non-synonymous site (dN, red) and the mean number of synonymous differences from the reference per synonymous site (dS, blue) for each of the three open reading frames (ORFs) core, pol, and S in the two groups for time point (**A**,**C**). Panel (**E**,**F**) display sliding window analyses of the mean number of pairwise non-synonymous differences per non-synonymous site (πN, red), as well as mean number of pairwise synonymous differences per synonymous site (πS, blue) for each of the three open reading frames (ORFs) core, pol and S in the two groups for time point (**A**,**C**).

**Figure 5 viruses-17-00684-f005:**
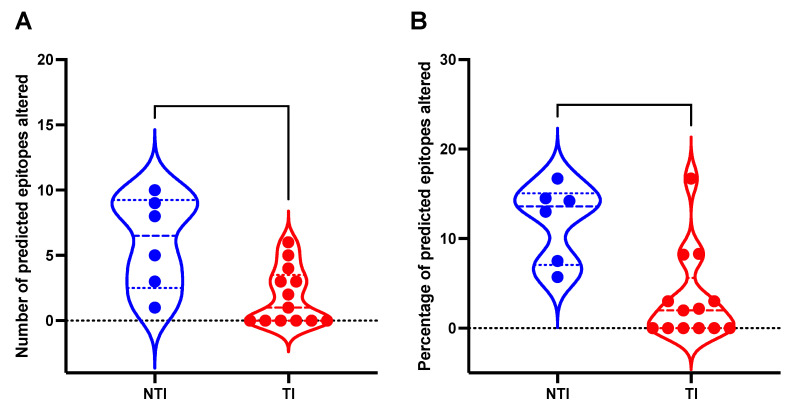
Mutations found in putative CD8^+^ T cell epitopes in 19 patients with Chronic Hepatitis B with and without treatment initiation. This figure depicts a violin plot with the distribution of (**A**) the number of mutations in predicted T cell epitopes in patients with treatment initiation (TI) and patients without treatment initiation (NTI), and (**B**) the fraction (%) of mutations in total predicted CD8^+^ T cell epitopes in the TI and NTI groups. Dashed lines inside the plot represent median value and the dotted lines represent the upper and lower quartiles.

**Figure 6 viruses-17-00684-f006:**
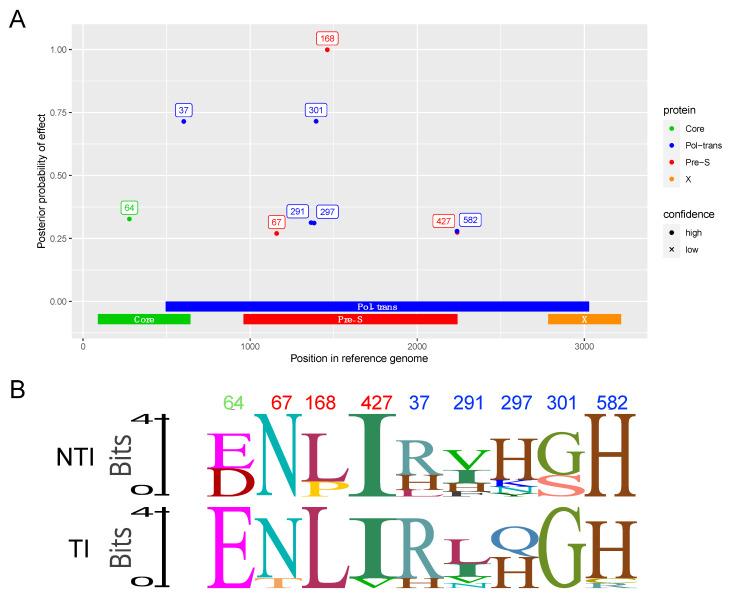
GWAS analysis of the Core, Pol-trans, Pre-S, and X-proteins encoded by the HBV genome. (**A**) Manhattan plot showing the strength of association between HBV genomic variants and the need for treatment. Points represent amino acid mutations, with the location in the genome (nucleotide numbering) shown on the x-axis and the y-axis showing how much support there is for this site having an impact on whether the patient needs treatment or not (the effect can be positive or negative). For the strongest signals, the location of the site in each protein is indicated on a label next to a point colored according to which protein the mutation was present in. Numbers on labels are amino acid numbering for that specific protein and correspond to locations in the protein sequence in the reference genome. Due to overlapping reading frames in the HBV genome, some DNA mutations have an effect on two different proteins. Analysis was based on amino acid sequences from 21 patients—7 where treatment was not necessary and 14 where treatment was needed. (**B**) Logo plot of the nine residues comparing the NTI group (top) to the TI group (below). Residue numbers are marked in color matching proteins listed in (**A**).

**Table 1 viruses-17-00684-t001:** Patient characteristics of 25 patients with Chronic Hepatitis B with or without treatment initiation. Data is presented as median (min-max) or *n* (%).

	Patients with Treatment Initiation *n* = 14	Patients Without Treatment Initiation *n* = 11	*p*-Values
Sex			
Female (%)	8 (57%)	5 (45%)	
Male (%)	6 (43%)	6 (55%)	*p* = 0.6951
Age, years median (range)	31 (21–59)	36 (30–62)	*p* = 0.2479
Country of origin			
European (%)	1 (7%)	0	
Asian (%)	13 (93%)	9 (82%)	
African (%)	0	2 (18%)	*p* = 0.1833
Follow up time, months (range)	60 (10–99)	57 (24–112)	*p* = 0.6982
HBeAg positive at baseline (%)	9 (64%)	0 (0%)	*p* = 0.001
Anti-HBe negative at baseline (%)	7 (50%)	0 (0%)	*p* = 0.0078
HBVDNA, IU/mL ^1^			
Sample A (range)	4.025 × 10^7^ (350–4.7 × 10^8^)	9000 (66–40,000)	
Sample B (range)	6.075 × 10^7^ (47,500–2.2 × 10^9^)	22,200 (10,500–31,500)	
Sample C (range)	2.366 × 10^7^ (13,000–1.7 × 10^11^)	15,100 (21–1.290 × 10^7^)	*p* = 1.3 × 10−14 *
ALT ^2^			
Sample A			
Normal (%)	9 (64.3%)	10 (91%)	
1×–2× ULN (%)	2 (14.3%)	1 (9%)	
>2× ULN (%)	3 (21.4%)	0 (0%)	
Sample B			
Normal (%)	12 (8.8%)	11 (100%)	
1×–2× ULN (%)	1 (7.1%)	0	
>2× ULN (%)	1 (7.1%)	0	
Sample C			
Normal (%)	8 (57.2%)	10 (91%)	
1×–2× ULN (%)	3 (21.4%)	0	
2× ULN (%)	3 (21.4%)	1 (9%)	*p* = 1.4 × 10^−6^ *

HBeAg: Hepatitis B e antigen. Anti-HBe: Antibodies against hepatitis B e antigen. ^1^ HBV DNA (hepatitis B virus DNA measured as international units/mL (IU/mL) in blood). ^2^ ALT (alanine aminotransferase) reference: 10–70 Units/L (male), 10–45 Units/L (female). UL N: (Upper Limit of Normal). ALT and HBV DNA values are from blood samples matched with the three sample time points. * *p*-values are based on results for ALT and HBV DNA registered in DANHEP during the study period.

## Data Availability

The raw data supporting the conclusions of this article will be made available by the authors on request.

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
