# Peer review of "Higher Rates of Viral Evolution in Chronic Hepatitis B Patients Linked to Predicted T Cell Epitopes"

_viruses, 2025, doi:10.3390/v17050684_

Round 1

Reviewer 1 Report

Comments and Suggestions for Authors

Comments

Dalegaard et al. examined substitution rates in patients with chronic hepatitis B virus infection who either were or were not subsequently administered nucleoside analog antiviral therapy. For each patient, three serial plasma samples were sequenced using MiSeq. Consensus sequences were aligned using MAFFT, and phylogenetic trees were generated using PhyML and FigTree. Substitution matrices were calculated using Geneious and analyzed usng SNP genie and transcoder. The HLA type of each patient was determined, and potential CD8+ T cell epitopes were predicted using the IEDB database. Finally, BMAGWA was used to identify substitutions associated with treatment initiation. This is a clever approach that seems to suggest that chronic HBV patients who do not require treatment tend to demonstrate stronger host immune pressure as observed through higher substitution rates in the HBV genome.

Major comments

  1. Although the hypothesis is sound and the conclusions appear to be reasonable and well-supported, it is a little difficult to follow the experimental design. The two experimental groups are somewhat artificial in terms of whether or not the patients was later administered antiviral therapy. This is interesting, especially in terms of identifying predictive factors for high and low-risk patients, but as the authors note, this decision is based to some extent on the discretion on the attending physician. To a large extent, the authors have instead classified the patients with respect to high and low HBV DNA levels (line 109), and perhaps they should focus on this more objective metric instead.
  2. The authors examined the substitution rates and the relationship with HBV DNA levels and the presence of HBe antigen and anti-HBe antibody in relation to T cell epitopes and immune evasion. The authors might also consider the potential role of APOBEC cytidine deaminases as an innate immune defense against HBV, which introduces mutations through C to U modifications. This might be detectable through examination of the signature surrounding mutation hotspots.
  3. It is interesting that the authors found no significant association between HBV genotype and the treated vs non-treated groups, but it might be useful to compare substitution rates with respect to HBV genotype. Fig. 2 shows clear clustering of within-patient samples, but there seems to be no consistent pattern with respect to changes over time (from time points A to B to C). Based on the study’s hypothesis, it is unclear if differences in the ordering or lengths of branches are expected.
  4. The study includes only patients who were treated with tenofovir or entecavir. This has no bearing on the study since only samples collected prior to treatment are used, but peg-interferon is also widely used to treat patients with chronic HBV, and the authors’ idea of classifying patients with respect to treatment might also be useful in predicting whether NA therapy or interferon therapy would be more effective for a given patient.

Minor comments

  1. There is a verb agreement error on lines 26-27: “Genome-wide association analysis revealed several amino acid residues in the HBV genome that was associated with treatment initiation.” This could be simplified to “Genome-wide association analysis revealed several amino acid residues in the HBV genome associated with treatment initiation.”
  2. Some background on the HBV genome is clearly necessary to introduce the concept overlapping reading frames, but the description around line 50 becomes a little convoluted. This section should focus more on the interesting quirk of the HBV genome that a single nucleotide change in a region of overlap can either have no effect or can alter the amino acid sequence of one or even both of the proteins and is thus exposed to unusual selection pressures.
  3. The dash on line 94 seems unnecessary: “indication, drug initiation and – cessation dates, and effect of treatment.”
  4. The contraction should be removed on line 223: “didn’t have corresponding HBVDNA values registered in the database”

  1. Line 243 should be rephrased: “*Statistic-values are based on all results for ALT “
  2. The formatting of the HBV DNA levels in Table 1 is confusing.
  3. Please rephrase line 336: “Using this approach, 9 sites that with high confidence were associated with treatment status was found.”

Reviewer 2 Report

Comments and Suggestions for Authors

The authors compared intra-individual HBV evolution over time in two groups of CHB patients in Denmark, using antiviral treatment initiation as an indicator of immunological control failure. The research idea is interesting and I agree that the development and selection of accurate, sensitive, and easily operable methods are crucial for HBV mutation research to shift towards guiding clinical diagnosis and treatment. Specific comments follow.

Major points:

  1. Please discuss the importance of sequencing the circulating HBV rather than HBV cccDNA in the infected cells.
  2. Figure 4: Please swap the color of “(πN, red)” and (πS, blue)” for consistency.
  3. Figures 3&5: Please explain horizontal bars.

Minor points:

  1. Line 88: Does this mean both CHB and CHC patients were enrolled?
